environmental science

capacity building, coffee agroforestry, community engagement, forest fragments, wildlife corridors

**Author for correspondence:**
Robin L. Chazdon
e-mail: rchazdon@usc.edu.au

One contribution to the special collection 'Sustainable land use: successful initiatives and state of art'.

# People, primates and predators in the Pontal: from endangered species conservation to forest and landscape restoration in Brazil's Atlantic Forest

Robin L. Chazdon[1], Laury Cullen Jr[2], Suzana M. Padua[2] and Claudio Valladares Padua[2]

[1]Tropical Forests and People Research Centre, University of the Sunshine Coast, 90 Sippy Downs Drive, Sippy Downs, Queensland 4556, Australia
[2]Instituto de Pesquisas Ecológicas, Rod. Dom Pedro I, km 47, Caixa Postal 47, Nazaré Paulista, SP, Brazil

  RLC, 0000-0002-7349-5687

This study describes the 35-year progression of activities in the Pontal do Paranapanema region of São Paulo State, Brazil. These activities began as a research project on the conservation ecology of the highly endangered Black Lion Tamarin and broadened into a landscape-scale restoration and conservation project involving the active participation of hundreds of landless families that colonized the region. Rather than viewing these colonists as a threat, a non-governmental organization arose to address their needs, providing training and support livelihoods. Local communities were engaged in conservation and restoration activities focused on studying the movement patterns of endangered species, environmental education programmes, planting native trees along riparian corridors, establishing coffee agroforestry plantings and initiating community-managed nurseries for the production of local native seedlings and non-native fruit trees. Farmers gained knowledge, income and food security, and developed a sense of ownership and shared responsibility for protecting wildlife, conserving forest fragments and restoring forests. Land sharing and restoring forest functions within an agricultural landscape matrix created new opportunities for people and endangered wildlife. We explore how key factors and partnerships critically influenced the landscape trajectory and conclude with lessons learned that may be relevant to sustainable landscape initiatives in other contexts.

# 1. Introduction: chronicling the trajectory of sustainable landscape initiatives

Achieving sustainable land use is a staged process, much like preparing a grand banquet. Preparing a banquet is a major undertaking, requiring quality ingredients, good recipes, a well-equipped kitchen and labour. Sometimes a head chef is in charge of a large kitchen staff, but banquets can also be pot-luck affairs with contributions from all who attend. Achieving sustainable landscapes requires a similar coordinated endeavour that goes through different stages of activity and adaptive management. Different sets of actors and stakeholders provide labour, guidance and ingredients, but they enjoy the meal together. In an ideal world, the landscape becomes a sustained banquet that provides food and meaningful livelihoods for local people while supporting and providing refuge for wildlife and myriad forms of biodiversity in diverse and dynamic habitats.

Sustainable landscape management exists largely in the conceptual realm, with few real-world cases [1,2]. Implementing sustainable landscape initiatives is a challenging task, requiring enabling factors that can take decades or more to put into place. Forest and landscape restoration (FLR) is a specific type of landscape-based initiative that aims to balance different land uses in ways that restore ecological functions and social benefits in landscapes within forest biomes that have become deforested and degraded [3]. Within a landscape, different types of land use, including commercial forestry and agriculture, can coexist in combination with practices to conserve and restore native forest ecosystems to achieve a more sustainable balance that enhances livelihood opportunities and contributes to better social and ecological outcomes [4,5].

Although holistic approaches such as FLR are highly appealing, they are very difficult to implement and sustain [6,7]. Landscape approaches develop over time and usually require a departure from past practices on the ground and within institutions that are traditionally oriented and equipped to achieve more limited goals [1]. More often, landscape initiatives originate as spatially or thematically focused projects or programmes that are confronted by challenges that require a landscape approach to resolve. These challenges often bring conflicts over land use and property rights. Responding adaptively, inclusively and creatively to challenges can ultimately bring transformative change to a landscape, based on a new operational paradigm.

This is the story of how FLR took form in the Pontal do Paranapanema in the state of São Paulo, Brazil [5]. This case study illustrates the 35-year progression of actions that transformed a narrowly focused conservation-oriented research project into a landscape-scale restoration project with multiple social and ecological outcomes. We describe how an initial focus on conservation ecology of the black lion tamarin (*Leontopithecus chrysopygus*) evolved in response to unpredictable challenges, through broadening the scope and participation in programmes and growing human and social capacity, leading to new opportunities and greater socio-environmental impacts [8]. A non-governmental organization arose in the region to provide leadership, stability, expertise and support throughout this process.

This progression provides one of the few examples around the world of the trajectory of a successful and sustainable forest and landscape restoration programme. This case study also illustrates how land sharing and restoring forest functions within an agricultural landscape matrix created new opportunities for people and endangered wildlife. Following the presentation of historical background and details of projects, actors, and actions implemented, we explore how key factors and partnerships critically influenced the landscape trajectory. We conclude with lessons learned from this particular context that may be relevant to sustainable landscape initiatives in other contexts (table 1).

# 2. History and geography of the Pontal do Parapanema region

The Pontal do Paranapanema region (figure 1) encompasses an area of 18 845 km$^2$ in the extreme west of São Paulo State (22°30′ S, 52°20′ W), Brazil [9]. It belongs to the administrative zone of the Presidente Prudente district, comprising 20 municipalities. It is the second-poorest region of São Paulo State. The region is bordered by the Parana River on the north and the Paranapanema River on the south. Native vegetation in the region is classified as Seasonal Semi-deciduous Forest, one of the most threatened vegetation types within the Atlantic Forest hotspot for global biodiversity conservation [10]. The region has a tropical wet-dry climate [11], with a hot and rainy season from October to March, and a dry period from April to September. Mean annual precipitation is 1341 mm and mean annual temperature is 24.1°C. The topography is characterized by open hills, with gentle slopes below

**Table 1.** Conservation and restoration projects and activities at Pontal do Paranapanema.

| project (year initiated) | activity | | | | | |
| --- | --- | --- | --- | --- | --- | --- |
| | environmental education in schools and settlements | technical workshops and training courses | community-managed tree nurseries | agroforestry buffer strips and home gardens | agroforestry islands | full or partial planting of native tree species |
| Black Lion Tamarin conservation (1982) | X | X | | | | |
| Lowland tapir conservation (1996) | | X | | | | |
| Cooperative conservation with MST settlers to protect forest fragments (1995) | X | X | | | | |
| Green Hug (1997) | X | X | X | X | | |
| Coffee with Forest (2001) | X | X | X | X | X | X |
| Landscape Detectives (1997) | X | X | | | | |
| Atlantic Forest Corridors (2006) | X | X | X | | X | X |
| A Good Pontal for All/Eco-Negotiations (2009) | X | X | | | | |

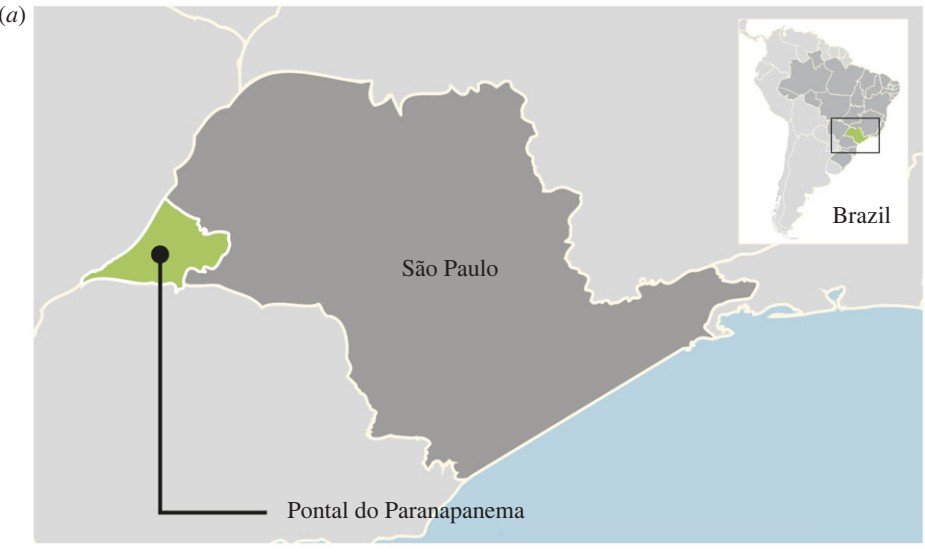

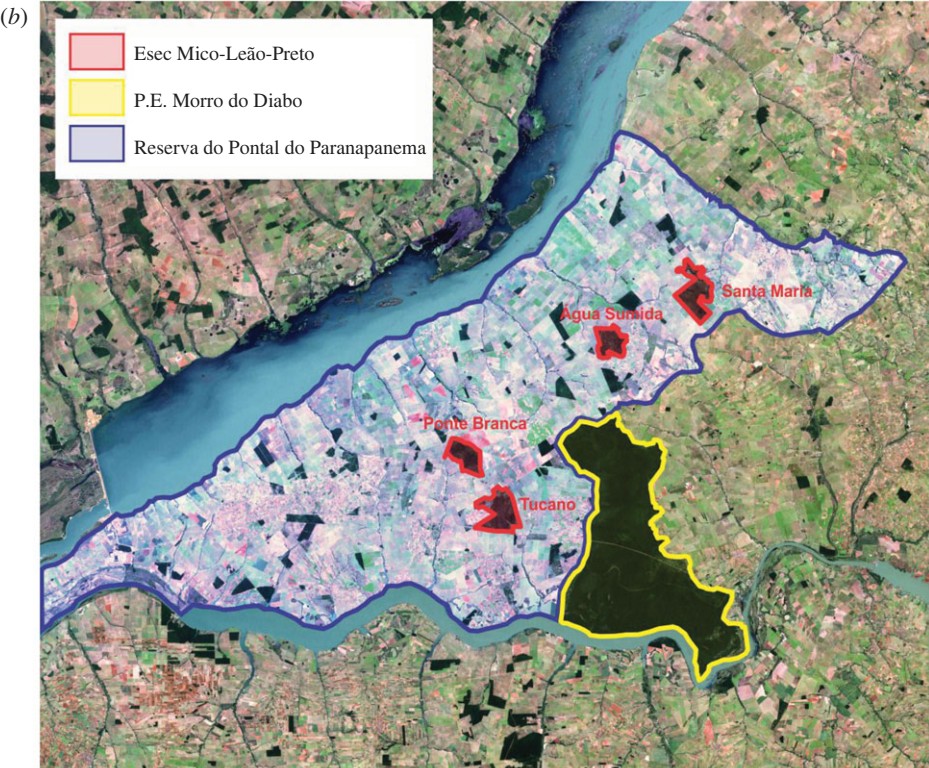

**Figure 1.** The location of the Pontal do Paranapanema region (*a*) and conservation units in 2000 (*b*). The large forested area outlined in yellow is the Morro do Diabo State Park. The four fragments that compose the Black Lion Tamarin Ecological Station are outlined in red.

15% [12]. The elevation in the region varies from 265 to 320 m, and predominant soil classes are the Ferrasols (Red Latosol) and Ultisols (Red-yellow Argisol) [13,14].

 The region remained largely forested in 1942 when three reserves were decreed by the state of São Paulo as refuges to protect the unique flora and fauna of this Atlantic Forest subregion. The largest was the 260 000 ha Great Pontal Reserve. The first wave of settlement in the area from 1945 to 1965—largely encouraged by the state government—led to deforestation of over 80% of the forest reserve to create large cattle ranching estates [15]. In 1986, most of the remaining forest of the Great Pontal Reserve was concentrated within a single large fragment of continuous forest of 36 000 ha, which became Morro do Diabo State Park, with another 13 200 ha distributed among isolated forest fragments [16] (figure 1 and electronic supplementary material, figure S1). Four of these fragments,

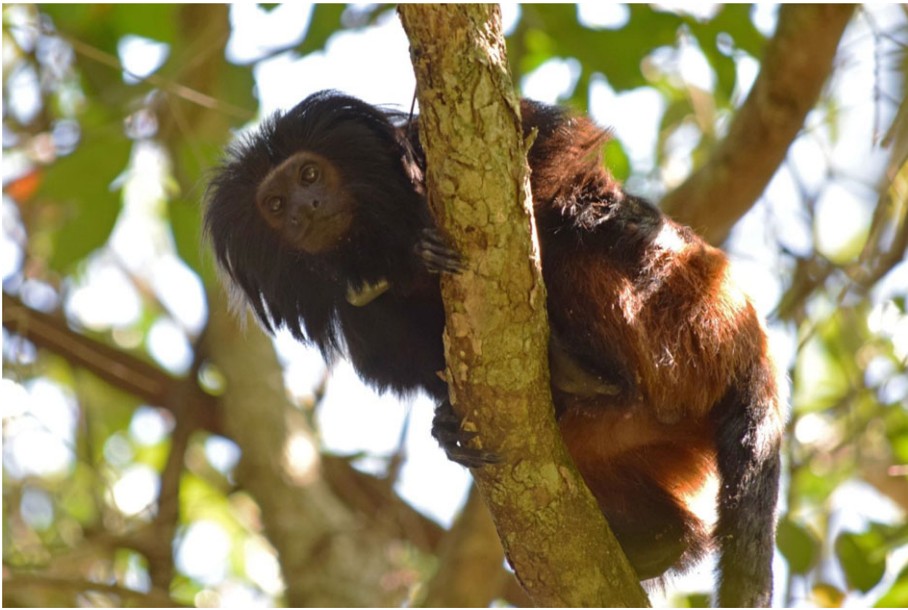

**Figure 2.** The black lion tamarin (Source: Wikipedia; photographer: Miguel Ranguel Jr).

with a total area of 6200 ha, became legally protected in 2002 when the Black Lion Tamarin Ecological Station was created by the federal government (figure 1). Another 7000 ha of forest fragments, ranging from 2 to 2000 ha, were scattered across the area on private land holdings. During the past decade, pasturelands on flat topography have been rapidly replaced by sugarcane plantations. In 1975, the state of São Paulo launched the construction of hydroelectric power plants on the Paranapanema and Paraná Rivers. Damming of the Paraná River began in the 1980s, flooding 3211 hectares, mostly of the Morro do Diabo State Park (electronic supplementary material, figure S1).

The Morro do Diabo State Park is the second largest forest remnant of Seasonal Semi-deciduous Forest in the Brazilian Atlantic Forest [17] of which only 1.8% of the original extent remains intact [18]. This region has one of the last populations of jaguar (*Panthera onca*) in the Atlantic Forest [19] and is one of the last refuges of the endangered black lion tamarin, one of the rarest New World primates [20]. The Pontal is also home to rare or threatened species of ocelots, pumas, tapirs, white-lipped peccaries, king vultures and the blue-and-yellow macaw [16].

## 3. Conservation biology in practice: shifting paradigms in human-modified landscapes

The black lion tamarin brought Claudio Valladares Padua to the Pontal in 1985. Also known as the golden-rumped lion tamarin, the black lion tamarin (figure 2) is endemic to the state of São Paulo and is one of the rarest of the New World monkeys. It was thought to be extinct for 65 years until it was rediscovered in 1970 by Coimbra Filho in Morro do Diabo State Park [20]. Claudio conducted his doctoral research on the conservation ecology of the black lion tamarin population within Morro do Diabo State Park, the only forest area in the region where the species was known to exist. Years later, it was also found in the Ecological Station of Caetetus, in the centre of São Paulo State, and more recently in many small fragments in the Morro do Diabo region and other areas of the State.

In 1990, Laury Cullen Jr graduated from the forestry school of the University of São Paulo and joined the research project as an intern at Pontal while Claudio returned to University of Florida to complete his doctorate. Another young researcher of Claudio's team, Eduardo Ditt [21] also joined the team. Laury began conducting surveys of forest fragments outside of the park, and over the next year he found eight groups of black lion tamarins using the fragments. This discovery changed the conservation approach for this species from a focus on a single population to a metapopulation focus. A metapopulation is a population of populations, or a group of groups, that is made up of the same species. Each subpopulation, or subgroup, is separated from all other subpopulations, but the movement of individuals from one population to another is needed for long-term survival of the metapopulation. The

Morro do Diabo's populations and captive-bred individuals were included in a viability plan to enhance long-term survival, consisting of a metapopulation strategy with translocations, reintroduction and managed dispersal among population groups. A population viability analysis based on a metapopulation scenario showed that the species could be saved if corridors were created and the population was managed at a landscape scale, including repopulating fragments from captive populations [20]. The species was classified as Critically Endangered by the IUCN species commission, with only 1000 known individuals (800 in the State Park and 200 in fragments outside the park). In fact, two populations have since been reintroduced into fragments where they were missing [22]. The conservation status of the black lion tamarin has since been downlisted from Critically Endangered to Endangered.

The realization that the entire landscape was the critical unit of conservation action for the black lion tamarin changed everything. This meant that, aside from focusing on protecting the species within Morro do Diabo State Park, work outside the park was needed to provide connectivity through biological corridors connecting with forest fragments. Outside the park were farmers and communities living in settlements. Their involvement in creating corridors would be essential.

Suzana Padua began her Masters degree in environmental education at the University of Florida while Claudio was a PhD student. She joined forces to initiate the Black Lion Tamarin Environmental Education programme in 1989. Focusing on school children ages 10 to 14 in grades 5–8, the programme was oriented toward providing environmental education to the people of the region who were underprivileged, both financially and educationally; only 50% had a third-grade education [23]. This age group was selected because they would be able to answer pre and post questionnaires.

The people living in areas surrounding Morro do Diabo State Park had few opportunities for learning. Educational materials and field trips focused on the black lion tamarin, providing an understanding of the animal's dependence on the forest and highlighting the importance of the interrelationships among species in forest ecosystems. The programme was successful in stimulating environmental awareness in the focal group of schoolchildren, teachers and overall communities. Community involvement first focused on raising people's awareness of the importance of protecting forest fragments and wildlife through social activities and fun events, such as music festivals and games. In addition, the programme created sustainable livelihood alternatives for small landowners, as a means to improve their lives, which were very impoverished. For example, the programme provided training on how to establish tree nurseries, how to plant particular tree species, as well as the production of handicrafts with nature themes. Over 400 families and thousands of urban dwellers of Teodoro Sampaio and vicinities participated in the programme. This effort was successful; during the two years of the programme's activities, there were no signs of degradation in the State Park [23].

Education became paramount from the very beginning, as interns joined the project and learned by doing with guidance from Claudio and Suzana Padua. Dozens of interns became involved, including many who are still actively engaged with programmes at Pontal, such as Laury Cullen Jr, Eduardo Ditt, Patricia Medici, Maria das Graças de Souza and Cristiana Martins. Interns were encouraged to pursue graduate studies in whichever field they thought they had talent, creating an interdisciplinary team of engaged, dedicated and passionate people.

When Claudio and Suzana returned to Brazil from their graduate studies, they planned to create an institution that could host conservation and education. Their plan materialized in 1992, when IPÊ (Institute for Ecological Research) was founded in Piracicaba, São Paulo State. Their idea was to form a partnership with the University of São Paulo Forest and Agriculture School in Piracicaba, to introduce the discipline of conservation biology, which did not yet exist in Brazil. The partnership did not succeed as expected, perhaps because the idea was too new and demanded interdisciplinarity at its core. This challenge inspired IPÊ to create its own education branch, now known as ESCAS, where it offers short courses, Masters and MBA degrees. Many of the student interns are still working with the IPÊ team today, and most teach as well as lead field projects. The lessons learned from the black lion tamarin project and other projects implemented over time guide what is taught and what needs attention. Additional researchers and interns joined IPÊ's team. The Institute is currently one of the largest environmental NGOs in Brazil and is recognized as a civic organization. It is now headquartered in Nazaré Paulista (São Paulo State) with a staff of over 90 professionals working on more than 40 projects throughout Brazil.

IPÊ uses an integrated action model, developed through years of experience, that combines research, environmental education, habitat restoration, community involvement with sustainable development, landscape conservation and policy-making (electronic supplementary material, figure S2). As a result, an efficient model was created in the Pontal that benefits not only humans but other forms of life. Its success relies on continuous, long-term involvement. This model has been used, with slight

modifications, in all regions where IPÊ works. A hallmark of IPÊ is its dedication to biodiversity conservation through science. The institution works with research, professional training, environmental education and sustainable income alternatives that emphasize socio-environmental responsibility from communities, corporations and opinion formers.

# 4. Agrarian reform meets conservation biology

The second wave of colonization in the Pontal began in July of 1990 when Brazil's Landless Workers Movement (MST in Portuguese) settled 800 families in the municipality of Teodoro Sampaio, adjacent to Morro do Diabo State Park [24]. Many of the large ranches granted to families as political favours in the 1950s and 1960s did not hold legal land titles, making this area an excellent opportunity for negotiation of land reform settlements. By 1995, over 6000 families occupied the area in 40 settlements [24]. In 1998, the MST families were awarded land tenure to the once private ranches. Land was removed from its original owners and redistributed in 14 ha farms among the families [25]. Under an arrangement with the São Paulo state government and landowners, the MST negotiated a deal in which landowners donate 30 to 70% of their cleared land to 'landless' families in exchange for official title to the remaining property. Much of the land that was donated to MST families was of marginal agricultural value bordering forest fragments and the Morro do Diabo State Park (figure 1).

In 1995, while working with the black lion tamarin conservation programme, Laury Cullen Jr began his Masters research in the Center for Latin American Studies and the Tropical Conservation and Development Program at the University of Florida. His research focused on assessing and reducing hunting pressure on deer, peccary and tapir in the region. Although park rangers were protecting wildlife inside the park, Laury heard frequent shooting in the forest fragments, evidence of high local hunting pressure. If actions were not taken quickly, these areas were going to become emptied of their wildlife.

Additional threats to the integrity of forest fragments were erosion of the forest edges by fires, vine colonization, wind desiccation, cattle grazing, spread of invasive plants and pesticide use [25]. Most of the landless families that occupied the Pontal region lacked local knowledge regarding farming and sustainable land use in lowland tropical conditions. About 20% came from urban areas and lacked agricultural skills [25]. How could IPÊ's team be solely focused on how to save endangered wildlife and prevent further degradation of forest fragments while people in the area were in such basic need? This concern led IPÊ to approach the MST leaders to work together to develop win–win solutions for people and nature. The result was an agreement to jointly plant trees in an effort to restore ecosystem functions following poor land use practices that caused soil depletion, lack of vegetation connectivity and shade, and water scarcity.

Laury then began to apply what he was learning at the university about community-based conservation, agroforestry and innovative approaches to conservation. He reasoned that creating buffer zones consisting of agroforestry strips of multipurpose trees and shrubs adjacent to forest fragments could accomplish two objectives at once: provide sources of useful forest products and reduce harmful 'edge effects'. In 1997, IPÊ initiated the 'Green Hug' project (Abraço Verde in Portuguese). A large grant from the Boticario Foundation provided technical assistance to 30 families living around forest fragments to establish agroforestry buffer strips to project forest edges, raise living standards and generate income on their land holdings. Farmers agreed to plant 60% of their seedlings along forest edges, with the remainder planted elsewhere on the farms or sold in local markets. On each farm, agroforestry was implemented in a strip of land bordering the edge of a forest fragment covering an area of approximately 2.5 ha, about 10–15% of the total farm area [25]. The buffer strips provided firewood, timber, fruits and fodder, reducing the need to harvest products or hunt animals in the forest fragments. Trees were interplanted with beans, maize and cassava, dietary staples. Families, self-organized in small groups, became involved in every stage of the project, from training and extension to project planning, implementation, monitoring and evaluation. Initial training and agroforestry extension were provided by short courses where community members learned and experienced the multiple benefits of agroforestry systems [25]. Today, 28% of the residents worked in agriculture before moving to a property in rural settlements.

Agroforestry plantings created a demand for seedlings, which initially had to be sourced from outside the landscape. In 2000, the first community nurseries were initiated in two settlements, Riberão Bonito and Tucano. Half of the species grown were native trees and half were *Eucalyptus* spp. By 2004 there were 14 community nurseries in the settlements, with a total of 102 families involved and a production capacity of 157 000 seedlings. All materials were provided by IPÊ, and families

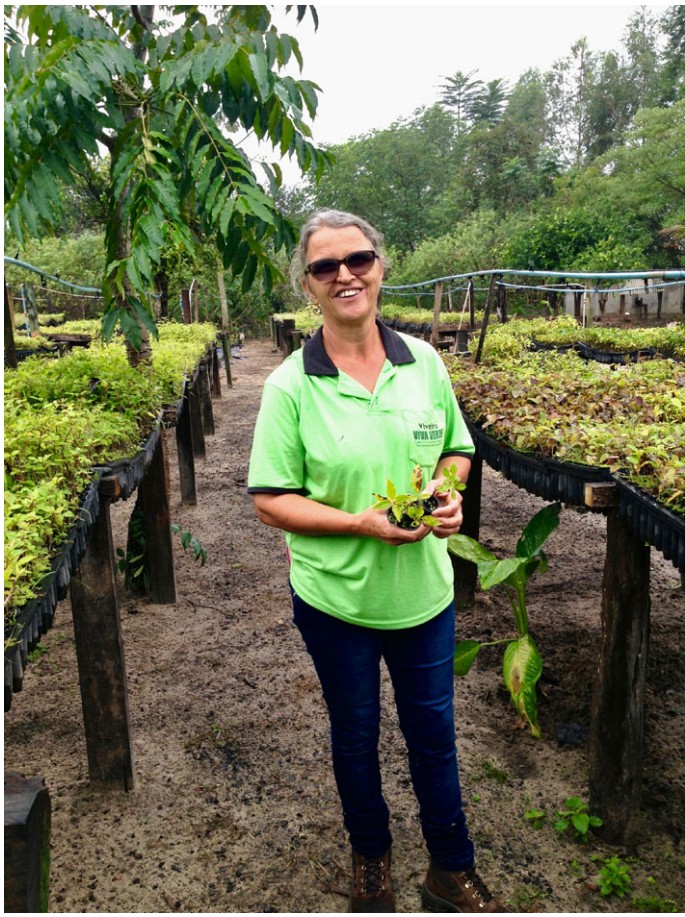

**Figure 3.** Iraci Lopes Duveza, owner of Viveiro Viva Verde (Green Life Nursery) for 15 years. 'Today, I see this as a business, the money invested comes from the IPÊ project, but it is also the result of my work, my investment.' (Photo by Robin Chazdon).

provided all labour. Women were responsible for managing the nurseries [26] (figure 3). Community members learned to identify and demarcate seed-bearing trees, plan seed-collection programmes and maintain viable seed [16]. These nurseries have also helped initiate an agroforestry culture in the region, stimulating the participation of other settlements [8,27].

Production agreements with local associations and families leading the nursery operations have generated approximately USD367 000 of local income from 2016 to 2019. Five local associations and five family-based operations participate in this process involving 115 people and 23 families from seed collecting to final seedling production. In 2019, the region had a total production capacity of about 885 000 seedlings per year. This labour brought a mean extra income of USD16 000/family or approximately USD450/month/family, considering a period of 36 months (3 years). Income data is collected from invoices and signed contracts with the nursery owner and reported in internal audits. The minimum wage, or Salário Mínimo, as it is known in Brazil, is the minimum amount set by the government for the salary of regular workers in Brazil, regardless of gender. The minimum wage (as of 1 January, 2016) is USD228 per month. The project contributes significant extra income per family corresponding to two minimum wages. In Pontal, the mean monthly income of a rural family is USD625. This means that the project provides an additional 60% of monthly income for participating families.

Improving agricultural productivity and establishment of community-based nurseries allowed local communities to (i) receive more income and build capacity to create new livelihoods, (ii) increase their awareness of environmental management of forest fragments and riparian forest, (iii) reduce the use of chemical pesticides, and (iv) become engaged in reforestation actions that broadened the buffer zone and helped to maintain the park and the fragments composing the Black Lion Tamarin Ecological Station.

The occupation by MST families initially posed a major challenge to the black lion tamarin conservation effort underway in the Pontal do Paranapanema. By integrating the settlers into landscape planning and providing training and capacity building to develop sustainable farming techniques, this challenge became an opportunity to adopt novel approaches through working

together. IPÊ promoted Eco-Negotiations, which are participatory forums where all segments of local society are encouraged to present their views about problems, potentials and viable solutions to the identified issues that can be solved when in partnerships and when the aim is to benefit the collectivity [27,28]. These meetings have been crucial to integrate and collectively plan the region's future. Many decisions and more than 20 projects have emerged and developed from these forums, and a number of conservation and agroforestry projects discussed and amply approved by local participants. Extensive and continual community education programmes, with input from engaged researchers, helps farmers and the overall public to understand the value of conservation and the potential of agroforestry to improve their quality of life.

# 5. Agroforestry stepping stones benefit families and wildlife

Forest and landscape restoration, when implemented in human-modified landscapes, is a land sharing process that integrates increases in tree cover through different types of interventions with economically beneficial activities [29]. Agroforestry systems provide an ideal approach, as they can be implemented using low chemical inputs and can be embedded within the landscape matrix in ways that provide habitat and resources for wildlife [30,31]. Further, agroforestry systems promote local traditional knowledge and can adapt to changing local markets and supply chains. In addition to buffer strips, agroforestry systems were also developed as islands along corridors connecting forest fragments to the park, providing additional benefits for wildlife that used these plantings to move around the landscape. Through the 'Coffee with Forests' programme initiated in 1997, IPÊ provided 38 families on small farms with technical training, and seedlings of coffee and tree species. Design of agroforestry systems for 'stepping stones' or intensive home gardens are planned at a minimum distance between lots to facilitate faunal movement and has incorporated shade-grown coffee from the start as a principal cash-generating component. These activities were designed to support the MST settlers, particularly those that lived around the large forest fragments that constitute the Black Lion Tamarin Ecological Station (figure 1).

Agroforestry islands based on organic shade-coffee production were established on many land holdings, providing income generation as well as natural 'stepping stones' to enhance wildlife movement and habitats within the areas surrounding forest fragments and the park (figure 4). Each planting is about 1 ha, a small orchard surrounding the family home. In turn, the families agreed to avoid using pesticides or industrial fertilizers and to use organic farming practices. Subsistence food crops, including papaya, banana, pumpkin, corn, cassava, cherry tomatoes, watermelon, cantaloupe, sweet potatoes, okra, rice and beans were planted between rows of trees and shade-coffee.

Since 1997, agroforestry has expanded to over 300 ha in the Pontal region, contributing to environmental conservation and sustainable rural development [32]. Today, 55 families are engaged in these productive systems covering approximately 70 ha in agroforestry stepping stones. In the productive sites, expected coffee production is 600 kg of processed coffee per hectare each year, with an income of up to USD1000 per hectare each year for landowners. In addition to providing modest income and food supplies for families, shade-coffee agroforestry plantings serve as 'stepping stones' in the landscape, allowing birds and insects to move around in the areas between larger forest fragments and the Morro do Diabo State Park. Although the agroforestry areas were not suitable habitats for forest-requiring bird species, they did provide ample resources for generalist bird species and contributed to the overall conservation of bird species and fruit-feeding forest butterfly species significantly more than prior monoculture agricultural practices [17,33]. Agroforestry areas studied in the Pontal provide rich resources for frugivorous birds, such as toucans, aracaris and macaws [34].

A recent study of ecological outcomes of 20 coffee agroforestry plantings found that after 15 years these agroforests had higher tree sapling abundance and canopy cover than restoration plantation of mixed native species established in the Pontal region at the same time [35]. For tree saplings, species density and the proportion of animal-dispersed species were higher in agroforests compared to restoration plantations. The management and high diversity of species planted in agroforests fostered the diverse natural regeneration of native tree species. Although tree species composition in agroforestry areas and restoration plantations may not reach levels observed in protected areas, these interventions demonstrate a clear trajectory of recovery. Further, in this region, agroforestry can be a cost-effective pathway for restoration that provides direct economic benefits for farmers, especially in the early years. Implementation costs for agroforestry and restoration plantations are around USD3000 and USD4800 per hectare, respectively [35].

(a)

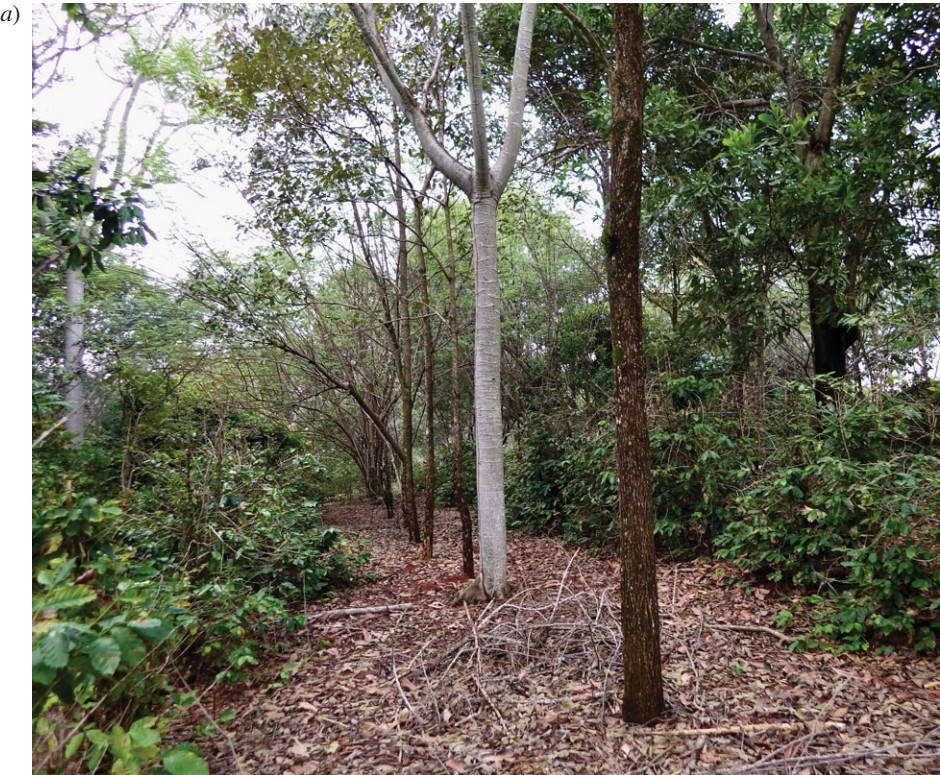

(b)

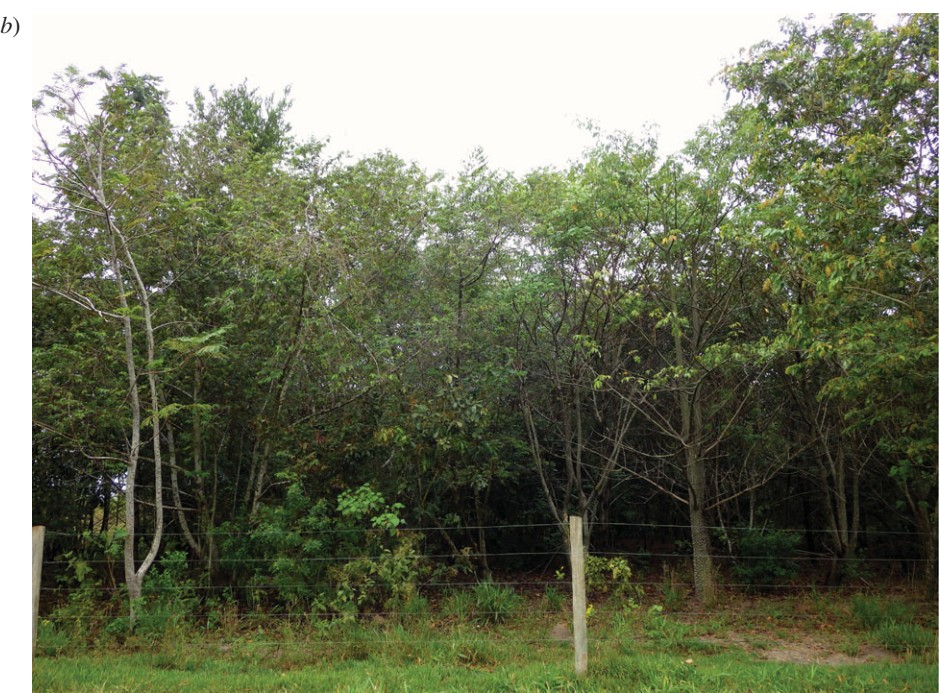

**Figure 4.** A coffee agroforestry 'stepping stone' in 2016, 11 years after establishment (*a*). Planted corridor of native trees on Rosanela Farm in 2016, two years after planting (*b*). (Photos by Robin Chazdon).

## 6. Reforesting corridors and landscape planning

When IPÊ began working with MST settlers to engage them in productive activities in support of conservation efforts in the region, it became evident that landscape-scale planning would be essential. The first plan involved planning a series of reforestation corridors that were mapped using the information on the movement of endangered wildlife throughout the area, including jaguars, pumas, ocelots, tapirs and the endemic black lion tamarins. The 'Landscape Detectives' project began in 2005

(a)

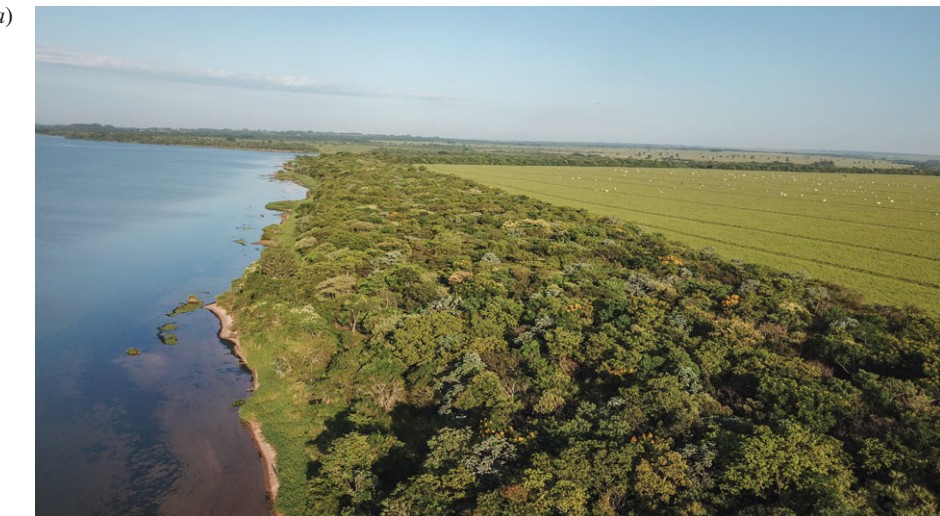

(b)

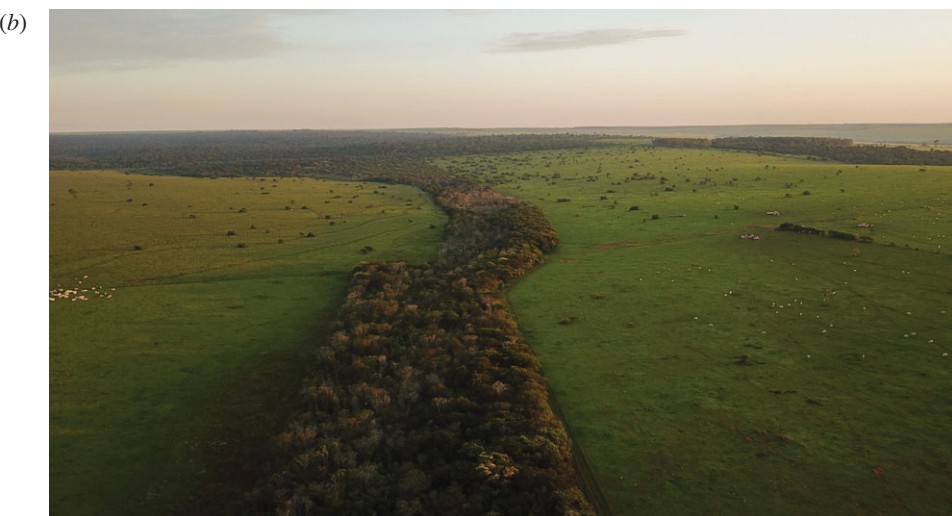

**Figure 5.** Aerial view of riparian corridor (*a*) and Rosanela corridor connecting riparian areas to Morro do Diabo State Park (*b*).

using radio-collared jaguars and pumas to identify strategic core areas and corridor areas to promote movement between suitable habitat patches. These movement patterns were used to generate a habitat map that identified important routes linking jaguar populations across the region. Population estimates based on radio-collared animals indicate that no more than 20 jaguars (*Panthera onca*), 30 pumas (*Puma concolor*), 120 ocelots (*Leopardus pardalis*) and 250 tapirs (*Tapirus terrestris*) survive in the Pontal do Paranapanema, levels considerably below the minimum viable size of 500 that is recommended for long-term survival of any species [20]. These findings confirmed that increasing landscape connectivity is a vital approach for conserving these threatened species.

Viewing the landscape from the animal's perspective was a key methodology used to develop a landscape plan involving reforestation of corridors linking core areas and wildlife reserves [18]. A key movement route was identified between the Morro do Diabo State Park and the Tucano forest fragment through privately owned Rosanela Farm. Laury Cullen was working for IPÊ on the Landscape Detectives project; in 2006, he approached the owner of Rosanela Farm, Vicente de Carvalho, about restoring farmland to develop this key wildlife corridor. As it turned out, Brazil's forest code obligated landowners to restore riparian zones to forest, so he became interested in the opportunity to collaborate. The owner provided fencing and land preparation, and IPÊ did the rest with support from a variety of sources including Petrobras, BNDES, Duke Energy, CTG Brazil, Whitley Fund for Nature, Durrell Wildlife Conservation Trust, Atvos, Natura, WeForest, Ecosia and many others from the voluntary carbon markets. Using a mix of 100 native tree species, 200 ha of the corridor were planted each year (figure 4). In 2010, IPÊ enacted a plan to reforest over 40 000 ha of land in forest corridors, connecting isolated forest fragments scattered across the landscape, while employing and providing livelihoods for over 1000 families. Rosanela Farm is now the location of the largest forest corridor ever planted in Brazil (figure 5).

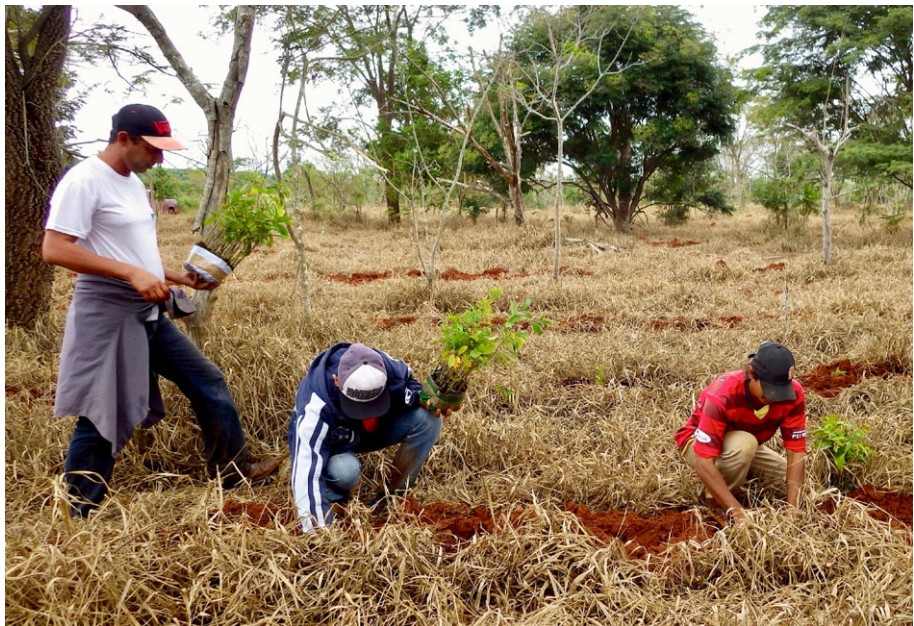

**Figure 6.** A team of community members planting seedlings in an assisted natural regeneration site treated with herbicides to kill invasive grasses. Photo by Robin Chazdon.

Using 3.0 million trees, and covering 1300 hectares of private land, this strip of land is already facilitating the movement of endangered wildlife between core zones of forest.

Besides enormous tree planting efforts for forest restoration in Rosanela Farm, natural regeneration also plays a key role for enriching, maintaining and expanding planted forests. At the Rosanela Farm, approximately 200 hectares are under assisted natural regeneration, where 18 native tree species were found regenerating in 2018. This less-intensive and less costly restoration activity generates income for local communities through the capacitation and hiring of local labour to carry out the many activities of assisted natural regeneration, such as controlling invasive grasses and leaf-cutting ants and fencing (figure 6). Natural regeneration sampled in forest restoration planting sites is composed of 50 species, of which 14 were not actively planted [35].

IPÊ's organizational model for wide-scale reforestation of the Atlantic Forest is encompassed in its 'dream map' for Pontal de Paranapanema. This conceptual map (figure 7) incorporates information on local rural properties, existing forest remnants, rural settlements and Private Protection Areas in riparian zones required by Brazil's forest code. The dream map is an approach to landscape planning, created by IPÊ and discussed with many stakeholders in the region during Eco-Negotiations, which are participatory meetings held at the Morro do Diabo State Park, or rarely in the public attorney headquarters in Presidente Prudente, the largest city in western São Paulo. The dream map is based on scientific assessments of water sources and rivers, the presence of endangered species in remaining habitats, along with information on who owns the land and where the priority areas for conservation are located. The goal is to maximize efforts on where to plant riparian forests, corridors and buffer zones, for example. All aspects are described publicly, so the understanding can lead to acceptance and social engagement in environmental issues. All of this information is pulled together to identify areas where reforestation efforts would be most beneficial and feasible. The dream map guided the creation of Brazil's largest reforestation corridor system, which after 10 years of effort, links two main remnants of Atlantic Forest in the Pontal de Paranapanema region, the Black Lion Tamarin Ecological Station and the Morro do Diabo State Park (figure 7).

From 2012 to 2018, reforested corridors have grown over 1500 ha (figures 5 and 7) neutralizing a net estimate of 156 000 Mg of $CO_2$ equivalents in the Pontal region after discounting for the baseline stocks of local pasturelands—where restoration is carried out—of 14.3 Mg $CO_2$e ha$^{-1}$ [36]. Local data on restoration sites indicate an annual gain of 12.8 Mg $CO_2$e ha$^{-1}$ of restoration during the first 5 years of restoration, with a potential to reach a total stock of 317.2 Mg $CO_2$e ha$^{-1}$ after 30 years [37]. By 2015 and 2017, a total of approximately 766 Mg of $CO_2$ equivalents were stored in aboveground biomass in the 122 hectares restored with support from WeForest [38]. Wildlife movements have also been documented using camera traps inside the corridors.

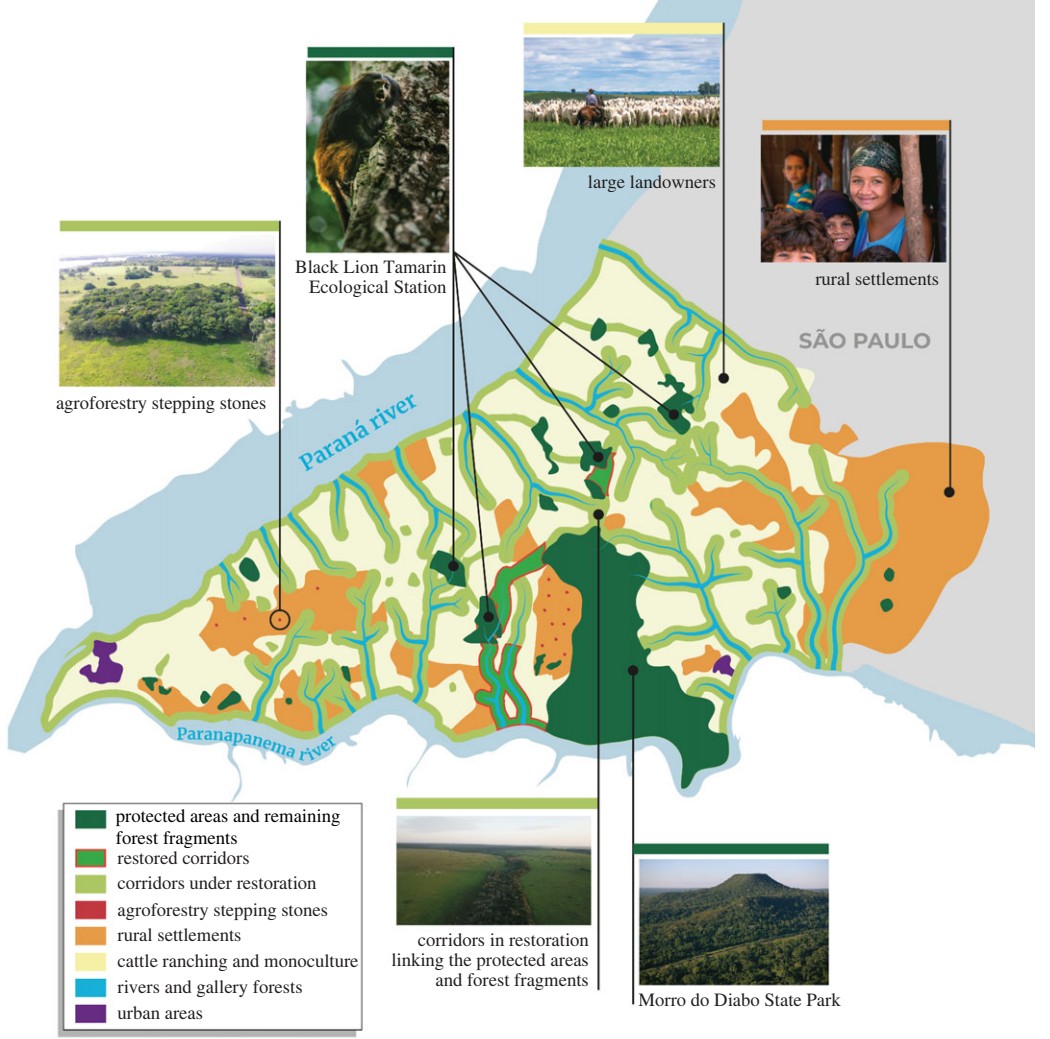

**Figure 7.** Dream map of the Pontal do Parapanema landscape. Reforested riparian corridors are outlined in red.

Over 35 years, what began as a research project on conservation ecology of the black lion tamarin gradually incorporated more aspects and goals [8,28]. As unanticipated challenges confronted the region, new programmes were developed, gradually building capacity of the people in local communities and of the NGO that emerged (table 1). This trajectory is guided by collaboration, communication, collective action and adaptive management, a feature that is emerging from other FLR case studies [7]. Projects became part of a long-term participatory process focused on rebuilding a landscape, connecting fragments, building community and providing opportunities for endangered wildlife and landless families. This process takes time and requires the participants to take ownership and responsibility. During the process, landless families acquired land and learned how to grow coffee, crops and trees. They became part of the landscape, sharing it with forest fragments, primates, jaguars and tapirs.

# 7. Key enabling factors and partnerships

Based on the history and experience of conservation and restoration in the Pontal do Paranapanema, a number of enabling factors emerged, including the development of several key partnerships and local institutional arrangements.

1. The activities conducted within this region became part of the sustained institutional mission of a non-governmental organization, IPÊ, dedicated to harmonizing the needs of the local communities with conservation and restoration practices. NGO leadership engaged in dialogues with MST settlers and large landowners to mediate conflicts and find solutions that provided economic benefits while

protecting and enhancing conditions for locally important wildlife. Individual projects were managed as part of a continuing long-term process.

2. Workshops and training of settlers were provided in partnership with government and agricultural agencies. This capacity building was essential for the development of agroforestry plantings and the establishment of local nurseries.

3. Collective action within the region was enabled by building long-term trust and developing a common vision for the landscape that recognized the needs of the local people for productive livelihoods and for engaged learning. These activities created a sense of community and collective purpose.

4. The local ownership and management of nurseries was fundamental to building capacity and developing livelihoods in support of the landscape conservation and restoration mission. The process of cultivating new trees from seeds harvested from the wild is owned entirely by local families who colonized the area as part of the landless rural workers' movement.

5. Agroforestry, conservation and restoration best practices were applied in the region based on scientific leadership and collaborations with national and international researchers. Pontal do Parapanema attracted the interest of interns and researchers in agroecology, environmental education, conservation and restoration, and also provided opportunities for community engagement in this research.

6. International companies and organizations have become engaged through projects offering carbon offset credits. By converting pastoral land into forest, huge amounts of carbon is absorbed from the atmosphere and locked away in vegetation and soils. These carbon offsets are quantified and monitored by the project, offering a unique way for programmes to both be financially sustainable and help groups meet their commitments towards combating climate change.

7. Environmental education programmes have been developed at all levels, from schools to community members of different segments of society, promoting the understanding of the importance of conservation and environmentally sound practices, which include local regional planning (Eco-Negotiations) to policies that help advance the ideas that benefit people and nature.

# 8. Conclusion and lessons learned

When forester Laury Cullen Jr first moved to Pontal do Paranapanema in São Paulo State to study an endangered primate species, 90% of Brazil's Atlantic Forest had been cleared at a devastating cost to the many hundreds of species of animals and plants living there, most of which are found nowhere else on Earth. Now, 30 years later, these devastating trends are slowly turning around in this corner of the world. Corridors and islands of forest are springing up anew, tended by the caring hands of 310 farming families who today earn a better living from the intermingled trees, wildlife and crops than they were ever able to gain from agriculture alone.

Laury's approach combined the goals of conservation and landscape restoration with finding new sources of income for poor farmers and funds for further forest replanting—a plan that earned him recognition as an Associate Laureate of the 2004 Rolex Awards. Today the scale of this achievement can be seen from space and via Google Earth, which shows the largest forest corridors (electronic supplementary material, figure S1). The corridors and associated 'stepping stones' contain 3 million trees comprising 1300 hectares of new forests that link two largest protected areas, the Black Lion Tamarin Ecological Station and the Morro do Diablo State Park (figures 1 and 7). For the first time in decades, the Mata Atlantica here is regrowing.

Looking back over 35 years of engagement with the Pontal do Paranapanema landscape and progression of activities, the following lessons provide insights into the outcomes and the new opportunities that have developed in the region. These lessons offer relevant guidance to work in other contexts and other places that aim to create more sustainable human-modified landscapes and align conservation goals with sustainable development.

1. Community-based education and conservation need to be offered on a continuous and long-term basis. Short-term projects, in isolation, are less effective for mobilizing change and building capacity.

2. Landscape planning at a regional scale is possible with the active participation of local communities. Local people and their farms are all part of the landscape and the contributions of all of these stakeholders synergize when there is a common plan and vision for the region. Public officials, such as the prosecutor (Ministério Público), also play a crucial role in informing and putting pressure on landowners to comply with mandated legal reserves and protected areas.

3. Collaboration with local families requires understanding their needs and meeting them where they are. Working to provide the tools that can respond to the needs of local families is the first step to integrating them into landscape-based conservation and restoration activities.

4. Promoting agroecology, adaptive management, adequate monitoring and appropriate policies can be critical to integrate individuals and groups into prioritizing measures beneficial to human and wildlife. Providing for the needs of people is an essential step towards reducing pressure on endangered wildlife and vegetation in forest fragments.

5. Sharing technical expertise in agricultural practices and environmental education allowed for collaborative planning of priority areas for wildlife corridors. Training and capacity building provides tools for local people to contribute to conservation and restoration practices.

6. Trust is an essential ingredient. When the trust level gets high enough, people transcend apparent limits, discovering new and awesome abilities of which they were previously unaware. When the trust account is high, communication is easy, instant and effective, which facilitates innovation and adoption of new systems.

7. The importance of working at the landscape scale is reinforced by using the landscape as a conservation biology and restoration ecology laboratory and testing ground.

8. It takes experience, time and effort to learn how to manage the equation of involving local communities in biodiversity conservation and forest restoration. In the Pontal, large landowners provided the land for forest restoration while the land reform settlers provide the services and the trees for the FLR programme. A local NGO working with the prosecutor (Ministério Público) can shoulder much of the burden of local governance and coordination across different stakeholders.

9. Viable landscapes can be rebuilt from small fragments through regional planning and with strong and broad support among landholders. Mapping and knowing how people and animals use the landscape permits effective conservation planning and identification of restoration priorities.

10. Institutional presence is key. Over time, as capacities and resources allow, new programmes and initiatives can develop, enabling greater participation and fine-tuning to the local context. IPÊ has been operating in the region for nearly 30 years safeguarding continuous financial support, community participation and adaptive management of a long-term FLR process. Sustainable conservation and restoration measures can be implemented by a team of integrated professionals that work together to respond to the complex needs of the real world.

Data accessibility. This article does not contain any additional data.

Authors' contributions. R.L.C. drafted the final version of manuscript; L.C.Jr, S.M.P. and C.V.P. provided data, historical information and maps. All authors made substantial contributions to the manuscript and approved the final version to be published. All authors agree to be accountable for all aspects of the work in ensuring that questions related to the accuracy or integrity of any part of the work are appropriately investigated and resolved.

Competing interests. The authors have no competing interests.

Funding. This study (including R.L.C.'s travel to the study area) was financed by the Coordenação de Aperfeiçoamento de Pessoal de Nível Superior – Brasil (CAPES) and WeForest Asbl/Vzw.

Acknowledgements. We thank Ricardo Gomes César and Mayte Benicio Rizek for providing additional data used in this case study. We thank Victoria Gutierrez and Pedro H. S. Brancalion for bringing us all together and the Institute of Ecological Research (IPÊ) and the staff of the Pontal Project for facilitating this study.

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
