## [Reviewer comments · Royal Society Open Science]

Review History

RSOS-200939.R0 (Original submission)

Review form: Reviewer 1

Is the manuscript scientifically sound in its present form?

Yes

Are the interpretations and conclusions justified by the results?

Yes

Is the language acceptable?

Yes

Do you have any ethical concerns with this paper?

No

Have you any concerns about statistical analyses in this paper?

No

Recommendation?

Accept with minor revision (please list in comments)

Comments to the Author(s)

This manuscript describes 35 years of outstanding conservation work in Pontal do Paranema, providing details on how from a research study the project grew into active participation from the local communities to transform the landscape for both humans and wildlife via agro-forestry and planting forest corridors. This important case study highlights the value of inclusive, landscape-scale approaches to conservation - which are becoming essential to ensure the safeguard of non-human primate populations, and indeed many other threatened species such as carnivores, as the majority of these taxa increasingly inhabit human-dominated and fragmented landscapes. The work at Pontal demonstrates the benefits of considering the conservation potential of human-influenced space, incorporating it into landscape-scale strategic approaches. The case study and conservation model presented here are highly relevant to a wide audience of conservation scientists and practitioners across the world.

The overall presentation of the case study could benefit from adding some more quantitative information, or in some cases to presented it earlier on, to help the reader visualise the progress over time. To avoid lengthy text, perhaps most of the quantitative information could be presented in the form of a table (similar to Table 1) for each section including number of families/villages involved and an estimate of the size of the area of intervention, divided into the different time stages.

Because of the large scale, I feel that the maps presented in SI do not do complete justice at showing the impact of the project (details get lost in the large scale). I suggest adding an insert to these maps, or even an additional Figure, "zooming in" to clearly show the evolution of some of the corridors from areal imagery, perhaps with the help of thin lines/borders using different colours for corridors and agro-forest areas to help the reader see the change in the structure of the landscape.

Specific comments:

Page 2, Line 27: Change to "black lion tamarin"

Page 4, Line 28: insert (FLR) after Forest and landscape restoration.

Page 4, Line 41: change citations to (6, 7).

Page 7, Figure 1: specify what the grey scale represents in the reference maps. Change "large-scale map" to "bottom map" or similar

Page 8, Figure 2: Is there a way you could merge this with Figure 1?

Page 8, Line 39: Is the species now known to occur elsewhere in the State of São Paulo? Because then in Page 9 Lines 9-12 it appears that they are only present in and around the Morro do Diabo State Park. And in Page 6 Line 43 you write "one of the last refugees" -> re-word this to "possibly the last known refugee" or similar if the species is not known to occur elsewhere, or specify that there are no population estimates outside your conservation area despite their known occurrence elsewhere (in Page 9).

Page 8, Line 45: specify where it was rediscovered (I assume in the Morro do Diabo State Park)? Consider moving the last two sentences ("Also known as the golden-rumped lion tamarin...rediscovered in 1970.") to the beginning of the paragraph and move "(Fig. 3)" after mentioning the species.

Page 9, Line 10: change "critically endangered" to "Critically Endangered". Change state park to State Park?

Page 9, Line 56: Is it possible to provide an average or approximate age range for the school children in grades 5-8?

Page 10, Line 8: Change to "black lion tamarin"

Page 10, Line 16: Could you provide more information regarding the groups of people involved in these activities, such as approximate numbers of villages and participants, and whether "overall communities" refers to indirect acquisition of environmental awareness or direct participation in activities

Page 10, Line 22: Who are the interns?

Page 10, Lines 25-26: "did not wish to leave the original group as they graduated" not clear. Also, in this paragraph it is not clear who the interns are, are they a different group of students to the University students or the schoolchildren?

Page 10, Lines 45: Again, does "original students" refer to University and/or local schoolchildren who then went to study at IPE/any university?

Page 12, Line 19: specify the area/width of the buffer zone or show in the map (Fig 1)

Page 12, Line 26: Should it be "Black Lion Tamarin Conservation Project"?

Page 12, Lines 38-41: The last sentence of this paragraph should be a new paragraph so that one paragraph refers to hunting and the next paragraph talks about land degradation.

Page 12, Lines 43-52: I would delete or reword "The settlers needed help and training, in many aspects. How could ... such basic need?"

Page 12, Lines 46 - 48: Reword sentence linking to the sentence ("Additional threats to the integrity of the forest... pesticide use.")

Page 12, Lines 49-52: This sentence needs rewording to something like: "The result was an agreement to plant trees in efforts to re-establish ecosystem functioning from land degradation that caused soil depletion, lack of vegetation connectivity and shade, and water scarcity". Or similar

Page 13, Line 15: Could you provide some quantitative information about number of farmers/villages involved in tree planting, and possibly some information about the spatial extent covered (again to represent the buffer zone, distance from the forest?) Maybe it would be helpful to use a table to show numbers of people/villages/areas involved over time.

Page 13, Line 31: Again, it would be interesting to add some information regarding number of farmers engaged across number of corridors and/or provide information on the extent of the area where activities took place, to help the reader visualise the scale and impact of the project.

Page 13, Line 43: Check if "Eucalyptus" should be italicised and include the species name

Page 14, Lines 5-7: instead of "during the past 36 months" use "between 20XX and 20XX (36 months)" or similar. In the next sentence start with "At present, .." ("At present, five local associations...")

Page 14, Lines 3-21: How were the economic data collected, was it through a survey? Conducted by? Add references where appropriate (eg. in "In Brazil the minimum wage... US\$228 per month" and perhaps specify if this minimum wage refers to all age/gender/occupation groups or if it is more specific). Check spelling for "US\$" across paragraph.

Page 14, Line 19: change "is providing" to "provides"

Page 14, Line 27: delete "to" so that it reads "...and 4) become..". Again, could you provide some quantitative evidence on how the buffer zone was "broadened", was the buffer zone pre-determined by mapped boundaries and the project helped restoring arboreal networks within it, or was it once narrow/non-existent and is now increased in width? Could you provide a map that can clearly show this?

Page 15, from Line 42: I feel like this part is similar to the section above (agro-forestry activities). I think the two sections should either be merged and words cut down (perhaps use a table to present numbers) or kept separate but re-worded and organised to make a clearer distinctions between the two sections.

Page 16, Line 10: "that enhanced wildlife movement". This statement needs evidence (i.e. reference(s) and an example), or reworded to "to enhance wildlife movement"

Page 18, Lines 10-17: Provide references for the population estimates and for the statement about the minimum viable size

Page 20, Lines 13-14: If possible, provide the full list of species (as SI) or is there a study published on this? If so, provide citations.

Page 20, Line 38: Do you mean "Figure 10"?

Page 21 Figure 10. Is there a way to show the reforestation progress in this map?

Page 21, Line 40: We Forest report should be a citation and included in the references.

Page 21, Line 45: "black lion tamarin"

Review form: Reviewer 2

Is the manuscript scientifically sound in its present form?

Yes

Are the interpretations and conclusions justified by the results?

Yes

Is the language acceptable?

Yes

Do you have any ethical concerns with this paper?

No

Have you any concerns about statistical analyses in this paper?

No

Recommendation?

Major revision is needed (please make suggestions in comments)

Comments to the Author(s)

This manuscript describes a case study, the Pontal do Paranapanema Forest and Landscape Restoration (FLR) project in Brazil, bringing also the lessons learned that may be relevant to initiatives in other contexts. This is an interesting story to tell. However, I have some comments aimed to improve manuscript presentation. Below, I detailed my comments:

General comments

To be placed in the “History...” or the “Conservation biology in practice:...” sections: How was the landscape delimited for this project? In FLR programs/project, delimiting the physical landscape is always something discussed and relevant. Knowing how it was done in that specific case would be very useful.

P9L3: briefly describe single population and metapopulation approaches for conservation. As the manuscript is not only for those working with this topic, a brief explanation is useful.

P9L41-P11L23. I really admire this successful history! However, I think this could be reduced in the manuscript, especially when it is not really connected to the Pontal do Paranapanema Project (example: the paragraph from P10L31-P11L5).

P14L14: Why don't you use the current Real to Dollar exchange rate? The current minimum wage is R\$ 1,045 and the exchange rate is currently 5.3 Real per Dollar. Thus, minimum wage is approximately US\$ 197. I think you should use this current exchange rate (2020) for all the conversations of Real to Dollar you have made in the manuscript.

It sounds strange to me the way some of the coauthors are mentioned in the text. For instance, Laury is a coauthor in the paper but in many parts he is mentioned in a way it seems he is outside it: P24L31: “Laury's approach combined... This occurs with other coauthors too. I am a non-native English speaker. Not sure if it is acceptable or not in English.

P20 – Paragraph starting L37. You mention a dream map and the idea of large forest corridors. Which is the goal in terms of forest cover and/or forest connectivity for the project's landscape? Was it discussed or established at any moment as a goal? Which have the project achieved in terms of forest cover increase in the landscape so far?

P21L35 – You could additionally state the mean annual increment of CO₂ or carbon per hectare per year by corridors. Moreover, instead of saying since 2006, you should state the right period. Is it from 2006 to 2020? Finally, I suppose you are mentioning the gross CO₂ uptake by forest restoration. I would not state this is CO₂ equivalents neutralized if you are neither discounting baseline (previous land use prior to reforestation) nor emissions during forest restoration implementation. If you are considering both discounts, ok. But add a brief sentence to make it clear.

Table 1: The caption could have more details and mention that Pontal do Paranapanema is in Brazil. Also, I think this table should not have horizontal lines, except in the borders. In addition, I think the deadline of each project could be added in case of projects that are no longer active. Also, a second column with project goal could be added. Finally, instead of only marking “X”, when available, you could put the goals and expected metrics in each activity for each project

Key enabling factors and partnerships:

1 – highlight that dialogues between MST settlers and large landowners are frequently hard and not easy to be conducted. If that was the case at Pontal, it is worth mentioning to increase the important of the IPÊ as a mediator for solving part of these conflicts.

6 – Is the carbon credits somehow verified by a third part in these processes (example VCS)? Who is the “owner” of this carbon offset? IPÊ?

Figures: I think figures could be overall improved and, when possible, those that were obtained from other sources (especially the maps) could be substituted or combined with figures prepared by the authors. Also, I think 10 figures are too much.

Figure 1, 2 and 10: They are showing basically the same landscape and information. If I understood, figures 1 and 2 were not produced by the authors. I encourage you to produce your own figures for the manuscript. In that case, I suggest merging figures 1, 2 and 10 in a colorful and informative figure 1. Another option is to merge 1 and 2 in a new figure, and put this new figure together with figure 10, that would be a single figure with panel A and B. This would show respectively the current/past landscape and the dream landscape. The mention of the protected areas (better to say protected areas instead of conservation units, as in table 2 caption) in 2 could be in figure 1. Names of the protected areas should be described in the figure caption. What does "C" mean in figure 1? Is it necessary to mention? If not, remove them. If yes, explain in the caption.

Figure 4 maybe could be a supplemental material.

Figure 5: The caption is too long. This could possibly be in the text, in the end of the paragraph starting P13L31. In addition, there is no citation to figure 5 in the text.

Figure 6: I suggest moving the citation of this figure in the text to the end of the sentence "Agroforestry islands based on...and the park (Fig. 6).

Figure 5-9. I suggest to put all of them in a single (or maybe two) figure with panels A-E.

Figure S1 - You should somehow indicate where the corridors are in the figure. At least, the most relevant ones should be pointed out.

Minor comments

P4L41: (6,7)

P5L12: provide authorship for *Leontopithecus chrysopygus*? If yes, do it for other names you mentioned.

P5L22: FLR program...

P6L31: you mean the São Paulo state? Or use state with the general meaning of country or nation?

P6L36: remove Supplemental Materials

P9L10: 1,000

P9L40: unit of conservation = area?

P10L12: use hectares instead of acres

P12L43: change the "." By "," before "many aspects".

P12L41: Ribeirão?

P14L15-16: to two minimum wages...

P17L26: forest restoration plantings or ecological restoration tree seedling plantings? Just to be clear it is about tree seedlings planted to restore tropical forests.

P21L35: 1,500

P21L37: Mg instead of tonnes?

P21L41: Cite the WeForest report?

P27L54: remove "22."

Decision letter (RSOS-200939.R0)

Dear Dr Chazdon,

The editors assigned to your paper ("People, primates, and predators in the Pontal: From endangered species conservation to forest and landscape restoration in Brazil's Atlantic Forest") have now received comments from reviewers. We would like you to revise your paper in

accordance with the referee and Associate Editor suggestions which can be found below (not including confidential reports to the Editor). Please note this decision does not guarantee eventual acceptance.

Please submit a copy of your revised paper before 12-Aug-2020. Please note that the revision deadline will expire at 00.00am on this date. If we do not hear from you within this time then it will be assumed that the paper has been withdrawn. In exceptional circumstances, extensions may be possible if agreed with the Editorial Office in advance. We do not allow multiple rounds of revision so we urge you to make every effort to fully address all of the comments at this stage. If deemed necessary by the Editors, your manuscript will be sent back to one or more of the original reviewers for assessment. If the original reviewers are not available, we may invite new reviewers.

- Data accessibility

<http://datadryad.org/submit?journalID=RSOS&manu=RSOS-200939>

- Competing interests

- Authors' contributions

All submissions, other than those with a single author, must include an Authors' Contributions section which individually lists the specific contribution of each author. The list of Authors

should meet all of the following criteria; 1) substantial contributions to conception and design, or acquisition of data, or analysis and interpretation of data; 2) drafting the article or revising it critically for important intellectual content; and 3) final approval of the version to be published.

- Acknowledgements

- Funding statement

on behalf of Dr Agnieszka Latawiec (Associate Editor) and Pete Smith (Subject Editor)
openscience@royalsociety.org

Comments to Author:

Reviewers' Comments to Author:

Reviewer: 1

Comments to the Author(s)

This manuscript describes 35 years of outstanding conservation work in Pontal do Parapanema, providing details on how from a research study the project grew into active participation from the local communities to transform the landscape for both humans and wildlife via agro-forestry and planting forest corridors. This important case study highlights the value of inclusive, landscape-scale approaches to conservation - which are becoming essential to ensure the safeguard of non-human primate populations, and indeed many other threatened species such as carnivores, as the majority of these taxa increasingly inhabit human-dominated and fragmented landscapes. The work at Pontal demonstrates the benefits of considering the conservation potential of human-influenced space, incorporating it into landscape-scale strategic approaches. The case study and conservation model presented here are highly relevant to a wide audience of conservation scientists and practitioners across the world.

The overall presentation of the case study could benefit from adding some more quantitative information, or in some cases to presented it earlier on, to help the reader visualise the progress over time. To avoid lengthy text, perhaps most of the quantitative information could be presented

in the form of a table (similar to Table 1) for each section including number of families/villages involved and an estimate of the size of the area of intervention, divided into the different time stages.

Because of the large scale, I feel that the maps presented in SI do not do complete justice at showing the impact of the project (details get lost in the large scale). I suggest adding an insert to these maps, or even an additional Figure, "zooming in" to clearly show the evolution of some of the corridors from areal imagery, perhaps with the help of thin lines/borders using different colours for corridors and agro-forest areas to help the reader see the change in the structure of the landscape.

Specific comments:

Page 2, Line 27: Change to "black lion tamarin"

Page 4, Line 28: insert (FLR) after Forest and landscape restoration.

Page 4, Line 41: change citations to (6, 7).

Page 7, Figure 1: specify what the grey scale represents in the reference maps. Change "large-scale map" to "bottom map" or similar

Page 8, Figure 2: Is there a way you could merge this with Figure 1?

Page 8, Line 39: Is the species now known to occur elsewhere in the State of São Paulo? Because then in Page 9 Lines 9-12 it appears that they are only present in and around the Morro do Diabo State Park. And in Page 6 Line 43 you write "one of the last refuges" -> re-word this to "possibly the last known refuge" or similar if the species is not known to occur elsewhere, or specify that there are no population estimates outside your conservation area despite their known occurrence elsewhere (in Page 9).

Page 8, Line 45: specify where it was rediscovered (I assume in the Morro do Diabo State Park)? Consider moving the last two sentences ("Also known as the golden-rumped lion tamarin....rediscovered in 1970.") to the beginning of the paragraph and move "(Fig. 3)" after mentioning the species.

Page 9, Line 10: change "critically endangered" to "Critically Endangered". Change state park to State Park?

Page 9, Line 56: Is it possible to provide an average or approximate age range for the school children in grades 5-8?

Page 10, Line 8: Change to "black lion tamarin"

Page 10, Line 16: Could you provide more information regarding the groups of people involved in these activities, such as approximate numbers of villages and participants, and whether "overall communities" refers to indirect acquisition of environmental awareness or direct participation in activities

Page 10, Line 22: Who are the interns?

Page 10, Lines 25-26: "did not wish to leave the original group as they graduated" not clear. Also, in this paragraph it is not clear who the interns are, are they a different group of students to the University students or the schoolchildren?

Page 10, Lines 45: Again, does "original students" refer to University and/or local schoolchildren who then went to study at IPE/any university?

Page 12, Line 19: specify the area/width of the buffer zone or show in the map (Fig 1)

Page 12, Line 26: Should it be "Black Lion Tamarin Conservation Project"?

Page 12, Lines 38-41: The last sentence of this paragraph should be a new paragraph so that one paragraph refers to hunting and the next paragraph talks about land degradation.

Page 12, Lines 43-52: I would delete or reword "The settlers needed help and training, in many aspects. How could ... such basic need?"

Page 12, Lines 46 - 48: Reword sentence linking to the sentence ("Additional threats to the integrity of the forest... pesticide use.")

Page 12, Lines 49-52: This sentence needs rewording to something like: "The result was an agreement to plant trees in efforts to re-establish ecosystem functioning from land degradation that caused soil depletion, lack of vegetation connectivity and shade, and water scarcity". Or similar

Page 13, Line 15: Could you provide some quantitative information about number of farmers/villages involved in tree planting, and possibly some information about the spatial extent covered (again to represent the buffer zone, distance from the forest?) Maybe it would be helpful to use a table to show numbers of people/villages/areas involved over time.

Page 13, Line 31: Again, it would be interesting to add some information regarding number of farmers engaged across number of corridors and/or provide information on the extent of the area where activities took place, to help the reader visualise the scale and impact of the project.

Page 13, Line 43: Check if "Eucalyptus" should be italicised and include the species name

Page 14, Lines 5-7: instead of "during the past 36 months" use "between 20XX and 20XX (36 months)" or similar. In the next sentence start with "At present, .." ("At present, five local associations...")

Page 14, Lines 3-21: How were the economic data collected, was it through a survey? Conducted by? Add references where appropriate (eg. in "In Brazil the minimum wage... US\$228 per month" and perhaps specify if this minimum wage refers to all age/gender/occupation groups or if it is more specific). Check spelling for "US\$" across paragraph.

Page 14, Line 19: change "is providing" to "provides"

Page 14, Line 27: delete "to" so that it reads "...and 4) become..". Again, could you provide some quantitative evidence on how the buffer zone was "broadened", was the buffer zone pre-determined by mapped boundaries and the project helped restoring arboreal networks within it, or was it once narrow/non-existent and is now increased in width? Could you provide a map that can clearly show this?

Page 15, from Line 42: I feel like this part is similar to the section above (agro-forestry activities). I think the two sections should either be merged and words cut down (perhaps use a table to present numbers) or kept separate but re-worded and organised to make a clearer distinctions between the two sections.

Page 16, Line 10: "that enhanced wildlife movement". This statement needs evidence (i.e. reference(s) and an example), or reworded to "to enhance wildlife movement"

Page 18, Lines 10-17: Provide references for the population estimates and for the statement about the minimum viable size

Page 20, Lines 13-14: If possible, provide the full list of species (as SI) or is there a study published on this? If so, provide citations.

Page 20, Line 38: Do you mean "Figure 10"?

Page 21 Figure 10. Is there a way to show the reforestation progress in this map?

Page 21, Line 40: We Forest report should be a citation and included in the references.

Page 21, Line 45: "black lion tamarin"

Reviewer: 2

Comments to the Author(s)

This manuscript describes a case study, the Pontal do Paranapanema Forest and Landscape Restoration (FLR) project in Brazil, bringing also the lessons learned that may be relevant to initiatives in other contexts. This is an interesting story to tell. However, I have some comments aimed to improve manuscript presentation. Below, I detailed my comments:

General comments

To be placed in the "History..." or the "Conservation biology in practice..." sections: How was the landscape delimited for this project? In FLR programs/project, delimiting the physical landscape is always something discussed and relevant. Knowing how it was done in that specific case would be very useful.

P9L3: briefly describe single population and metapopulation approaches for conservation. As the manuscript is not only for those working with this topic, a brief explanation is useful.

P9L41-P11L23. I really admire this successful history! However, I think this could be reduced in the manuscript, especially when it is not really connected to the Pontal do Paranapanema Project (example: the paragraph from P10L31-P11L5).

P14L14: Why don't you use the current Real to Dollar exchange rate? The current minimum wage is R\$ 1,045 and the exchange rate is currently 5.3 Real per Dollar. Thus, minimum wage is approximately US\$ 197. I think you should use this current exchange rate (2020) for all the conversations of Real to Dollar you have made in the manuscript.

It sounds strange to me the way some of the coauthors are mentioned in the text. For instance, Laury is a coauthor in the paper but in many parts he is mentioned in a way it seems he is outside it: P24L31: "Laury's approach combined... This occurs with other coauthors too. I am a non-native English speaker. Not sure if it is acceptable or not in English.

P20 - Paragraph starting L37. You mention a dream map and the idea of large forest corridors. Which is the goal in terms of forest cover and/or forest connectivity for the project's landscape? Was it discussed or established at any moment as a goal? Which have the project achieved in terms of forest cover increase in the landscape so far?

P21L35 - You could additionally state the mean annual increment of CO₂ or carbon per hectare per year by corridors. Moreover, instead of saying since 2006, you should state the right period. Is it from 2006 to 2020? Finally, I suppose you are mentioning the gross CO₂ uptake by forest restoration. I would not state this is CO₂ equivalents neutralized if you are neither discounting baseline (previous land use prior to reforestation) nor emissions during forest restoration

implementation. If you are considering both discounts, ok. But add a brief sentence to make it clear.

Table 1: The caption could have more details and mention that Pontal do Paranapanema is in Brazil. Also, I think this table should not have horizontal lines, except in the borders. In addition, I think the deadline of each project could be added in case of projects that are no longer active. Also, a second column with project goal could be added. Finally, instead of only marking "X", when available, you could put the goals and expected metrics in each activity for each project

Key enabling factors and partnerships:

1 - highlight that dialogues between MST settlers and large landowners are frequently hard and not easy to be conducted. If that was the case at Pontal, it is worth mentioning to increase the important of the IPÊ as a mediator for solving part of these conflicts.

6 - Is the carbon credits somehow verified by a third part in these processes (example VCS)? Who is the "owner" of this carbon offset? IPÊ?

Figures: I think figures could be overall improved and, when possible, those that were obtained from other sources (especially the maps) could be substituted or combined with figures prepared by the authors. Also, I think 10 figures are too much.

Figure 1, 2 and 10: They are showing basically the same landscape and information. If I understood, figures 1 and 2 were not produced by the authors. I encourage you to produce your own figures for the manuscript. In that case, I suggest merging figures 1, 2 and 10 in a colorful and informative figure 1. Another option is to merge 1 and 2 in a new figure, and put this new figure together with figure 10, that would be a single figure with panel A and B. This would show respectively the current/past landscape and the dream landscape. The mention of the protected areas (better to say protected areas instead of conservation units, as in table 2 caption) in 2 could be in figure 1. Names of the protected areas should be described in the figure caption. What does "C" mean in figure 1? Is it necessary to mention? If not, remove them. If yes, explain in the caption.

Figure 4 maybe could be a supplemental material.

Figure 5: The caption is too long. This could possibly be in the text, in the end of the paragraph starting P13L31. In addition, there is no citation to figure 5 in the text.

Figure 6: I suggest moving the citation of this figure in the text to the end of the sentence "Agroforestry islands based on...and the park (Fig. 6).

Figure 5-9. I suggest to put all of them in a single (or maybe two) figure with panels A-E.

Figure S1 - You should somehow indicate where the corridors are in the figure. At least, the most relevant ones should be pointed out.

Minor comments

P4L41: (6,7)

P5L12: provide authorship for *Leontopithecus chrysopygus*? If yes, do it for other names you mentioned.

P5L22: FLR program...

P6L31: you mean the São Paulo state? Or use state with the general meaning of country or nation?

P6L36: remove Supplemental Materials

P9L10: 1,000

P9L40: unit of conservation = area?

P10L12: use hectares instead of acres

P12L43: change the "." By ",," before "many aspects".

P12L41: Ribeirão?

P14L15-16: to two minimum wages...

P17L26: forest restoration plantings or ecological restoration tree seedling plantings? Just to be clear it is about tree seedlings planted to restore tropical forests.

P21L35: 1,500

P21L37: Mg instead of tonnes?

P21L41: Cite the WeForest report?
P27L54: remove "22."

Author's Response to Decision Letter for (RSOS-200939.R0)

See Appendix A.

Decision letter (RSOS-200939.R1)

Dear Dr Chazdon,

It is a pleasure to accept your manuscript entitled "People, primates, and predators in the Pantanal: From endangered species conservation to forest and landscape restoration in Brazil's Atlantic Forest" in its current form for publication in Royal Society Open Science.

on behalf of Dr Agnieszka Latawiec (Associate Editor) and Pete Smith (Subject Editor)
openscience@royalsociety.org

Appendix A

Comments to Author:

Reviewers' Comments to Author:

Reviewer: 1

Comments to the Author(s)

This manuscript describes 35 years of outstanding conservation work in Pontal do Parapanema, providing details on how from a research study the project grew into active participation from the local communities to transform the landscape for both humans and wildlife via agro-forestry and planting forest corridors. This important case study highlights the value of inclusive, landscape-scale approaches to conservation - which are becoming essential to ensure the safeguard of non-human primate populations, and indeed many other threatened species such as carnivores, as the majority of these taxa increasingly inhabit human-dominated and fragmented landscapes. The work at Pontal demonstrates the benefits of considering the conservation potential of human-influenced space, incorporating it into landscape-scale strategic approaches. The case study and conservation model presented here are highly relevant to a wide audience of conservation scientists and practitioners across the world.

Thank you for your positive comments.

The overall presentation of the case study could benefit from adding some more quantitative information, or in some cases to presented it earlier on, to help the reader visualise the progress over time. To avoid lengthy text, perhaps most of the quantitative information could be presented in the form of a table (similar to Table 1) for each section including number of families/villages involved and an estimate of the size of the area of intervention, divided into the different time stages.

We have added quantitative information where we could. We do not think a Table would be appropriate given that we don't have repeated data gathered at different time stages.

Because of the large scale, I feel that the maps presented in SI do not do complete justice at showing the impact of the project (details get lost in the large scale). I suggest adding an insert to these maps, or even an additional Figure, "zooming in" to clearly show the evolution of some of the corridors from areal imagery, perhaps with the help of thin lines/borders using different colours for corridors and agro-forest areas to help the reader see the change in the structure of the landscape.

It is true that the google earth images do not convey changes over time with much detail. Even after zooming-in there would not be much visible change. At this point we are not planning to do a detailed quantitative analysis of land cover change in this region using these images. Figure 10 shows where riparian corridors have already been reforested. If you don't think the SI figure is useful, that is fine. It can be deleted.

Specific comments:

Page 2, Line 27: Change to "black lion tamarin"

DONE

Page 4, Line 28: insert (FLR) after Forest and landscape restoration.

DONE

Page 4, Line 41: change citations to (6, 7).

DONE

Page 7, Figure 1: specify what the grey scale represents in the reference maps. Change "large-scale map" to "bottom map" or similar

Figure 1 has been replaced as much of this information is shown in the top part of Figure 1.

Page 8, Figure 2: Is there a way you could merge this with Figure 1?

Figure 1 has been replaced as much of this information is shown in the top part of Figure 1.

Page 8, Line 39: Is the species now known to occur elsewhere in the State of São Paulo? Because then in Page 9 Lines 9-12 it appears that they are only present in and around the Morro do Diabo State Park. And in Page 6 Line 43 you write "one of the last refugees" → re-word this to "possibly the last known refugee" or similar if the species is not known to occur elsewhere, or specify that there are no population estimates outside your conservation area despite their known occurrence elsewhere (in Page 9).

We added more information. The paragraph now reads: The black lion tamarin brought Claudio Valladares Padua to the Pontal in 1985. Also known as the golden-rumped lion tamarin, the black lion tamarin (Fig. 3), is endemic to the state of São Paulo and is one of the rarest of the New World monkeys. It was thought to be extinct for 65 years until it was rediscovered in 1970 by Coimbra Filho in Morro do Diabo State Park (20). Claudio conducted his doctoral research on the conservation ecology of the black lion tamarin population within Morro do Diabo State Park, the only forest area in the region where the species was known to exist. Years later it was also found in the Ecological Station of Caetetus, in the center of São Paulo State, and more recently in many small fragments both in the Morro do Diabo region and other areas of the State, where its original distribution was known to be.

Page 8, Line 45: specify where it was rediscovered (I assume in the Morro do Diabo State Park)? Consider moving the last two sentences ("Also known as the golden-rumped lion tamarin....rediscovered in 1970.") to the beginning of the paragraph and move "(Fig. 3)" after mentioning the species.

Done. See paragraph above

Page 9, Line 10: change "critically endangered" to "Critically Endangered". Change state park to State Park?

Done.

Page 9, Line 56: Is it possible to provide an average or approximate age range for the school children in grades 5-8?

Added ages 10-14.

Page 10, Line 8: Change to "black lion tamarin"

Done

Page 10, Line 16: Could you provide more information regarding the groups of people involved in these activities, such as approximate numbers of villages and participants, and whether "overall communities" refers to indirect acquisition of environmental awareness or direct participation in activities

Modified to: "The program was successful in stimulating environmental awareness in the focal group of schoolchildren, teachers, and overall communities. Community involvement first focused on raising people's awareness to the importance of protecting the remnants of forests and wildlife, through social activities and fun events, such as music festivals and games. In addition, the program created sustainable livelihood alternatives for small landowners, as a means to improve their lives, which were very impoverished. For example, the program provided training on how to establish tree nurseries, how to plant particular tree species, as well as production of handicrafts with nature themes. Over 400 families and thousands of urban dwellers of Teodoro Sampaio and vicinities participated in the program."

Page 10, Line 22: Who are the interns?

Added: "Dozens of interns became involved, including many who are still actively engaged with programs at Pontal, such as Laury Cullen Jr., Eduardo Ditt, Patricia Medici, Maria das Graças de Souza, and Cristiana Martins."

Page 10, Lines 25-26: "did not wish to leave the original group as they graduated" not clear. Also, in this paragraph it is not clear who the interns are, are they a different group of students to the University students or the schoolchildren?

Reworded: "Many of the student interns are still working with the IPÊ team today, and most teach as well as lead field projects."

Page 10, Lines 45: Again, does "original students" refer to University and/or local schoolchildren who then went to study at IPE/any university?

The interns were university students who had not yet begun graduate school. Many went on to do graduate studies at IPE or other universities.

Page 12, Line 19: specify the area/width of the buffer zone or show in the map (Fig 1)

Added: Agroforestry was implemented in an area of approximately 2.5 ha, about 10-15% of the total land area of each farm in a strip of land bordering the edge of a forest fragment.

Page 12, Line 26: Should it be "Black Lion Tamarin Conservation Project"?

Yes, corrected.

Page 12, Lines 38-41: The last sentence of this paragraph should be a new paragraph so that one paragraph refers to hunting and the next paragraph talks about land degradation.

This sentence has been moved to begin the next paragraph.

Page 12, Lines 43-52: I would delete or reword "The settlers needed help and training, in many aspects. How could ... such basic need?"

Deleted "The settlers needed help and training.."

Page 12, Lines 46 - 48: Reword sentence linking to the sentence ("Additional threats to the integrity of the forest... pesticide use.")

Done

Page 12, Lines 49-52: This sentence needs rewording to something like: "The result was an agreement to plant trees in efforts to re-establish ecosystem functioning from land degradation that caused soil depletion, lack of vegetation connectivity and shade, and water scarcity". Or similar

Now reads: The result was an agreement to jointly plant trees in an effort to restore ecosystem functions following poor land-use practices that caused soil depletion, lack of vegetation connectivity and shade, and water scarcity.

Page 13, Line 15: Could you provide some quantitative information about number of farmers/villages involved in tree planting, and possibly some information about the spatial extent covered (again to represent the buffer zone, distance from the forest?) Maybe it would be helpful to use a table to show numbers of people/villages/areas involved over time.

Added: The project initially provided technical assistance to 30 families living around forest fragments to establish agroforestry buffer strips to project forest edges, raise living standards, and generate income on their land holdings.

Page 13, Line 31: Again, it would be interesting to add some information regarding number of farmers engaged across number of corridors and/or provide information on the extent of the area where activities took place, to help the reader visualise the scale and impact of the project.

More data are included in this paragraph now.

Page 13, Line 43: Check if "Eucalyptus" should be italicised and include the species name

The species names are not available. Probably several species were used.

Page 14, Lines 5-7: instead of "during the past 36 months" use "between 20XX and 20XX (36 months)" or similar. In the next sentence start with "At present, .." ("At present, five local associations...")

Changed to 2016-2019

Page 14, Lines 3-21: How were the economic data collected, was it through a survey? Conducted by? Add references where appropriate (eg. in "In Brazil the minimum wage... US\$228 per month" and perhaps specify if this minimum wage refers to all

age/gender/occupation groups or if it is more specific). Check spelling for "US\$" across paragraph.

Changed all currency data to USD. The data were provided by Laury Cullen and a social scientist who works with the Pontal Project. I don't know the precise methodology used.

Page 14, Line 19: change "is providing" to "provides"

Done

Page 14, Line 27: delete "to" so that it reads "...and 4) become..". Again, could you provide some quantitative evidence on how the buffer zone was "broadened", was the buffer zone pre-determined by mapped boundaries and the project helped restoring arboreal networks within it, or was it once narrow/non-existent and is now increased in width? Could you provide a map that can clearly show this?

Done. The buffer zones are not mapped other than what is indicated in Figure 10. They reduced edge effects in the forest fragments.

Page 15, from Line 42: I feel like this part is similar to the section above (agro-forestry activities). I think the two sections should either be merged and words cut down (perhaps use a table to present numbers) or kept separate but re-worded and organised to make a clearer distinctions between the two sections.

This section is about the coffee agroforestry stepping stones, which are different from the buffer strips located at edges of forest fragments. I think the distinction is clear.

Page 16, Line 10: "that enhanced wildlife movement". This statement needs evidence (i.e. reference(s) and an example), or reworded to "to enhance wildlife movement"

Reworded

Page 18, Lines 10-17: Provide references for the population estimates and for the statement about the minimum viable size

added

Page 20, Lines 13-14: If possible, provide the full list of species (as SI) or is there a study published on this? If so, provide citations.

The list of species is provided in Supplemental data from another publication (Badari et al. 2020) so this citation is added.

Page 20, Line 38: Do you mean "Figure 10"?

Yes, changed. Thank you for noticing this.

Page 21 Figure 10. Is there a way to show the reforestation progress in this map?

It is not very visible yet on a map of the whole area.

Page 21, Line 40: We Forest report should be a citation and included in the references.

Added a citation to this report.

Page 21, Line 45: "black lion tamarin"

Changed to lower case.

Reviewer: 2

Comments to the Author(s)

This manuscript describes a case study, the Pontal do Paranapanema Forest and Landscape Restoration (FLR) project in Brazil, bringing also the lessons learned that may be relevant to initiatives in other contexts. This is an interesting story to tell. However, I have some comments aimed to improve manuscript presentation. Below, I detailed my comments:

General comments

To be placed in the "History..." or the "Conservation biology in practice:..." sections: How was the landscape delimited for this project? In FLR programs/project, delimiting the physical landscape is always something discussed and relevant. Knowing how it was done in that specific case would be very useful.

This did not originate as an FLR project so boundaries were largely historical due to the original Great Pontal Reserve, the State Park, and the key forest fragments.

P9L3: briefly describe single population and metapopulation approaches for conservation. As the manuscript is not only for those working with this topic, a brief explanation is useful.

New text added: A metapopulation is a population of populations, or a group of groups, that is made up of the same species. Each subpopulation, or subgroup, is separated from all other subpopulations, but movement of individuals from one population to another is needed for long-term survival of the metapopulation.

P9L41-P11L23. I really admire this successful history! However, I think this could be reduced in the manuscript, especially when it is not really connected to the Pontal do Paranapanema Project (example: the paragraph from P10L31-P11L5).

We disagree. These details are important to understand how the NGO was formed and how Claudio and Suzana created opportunities for interns to become engaged in research and training. This is part of the development of human and social capital over time that depends on leadership and involvement of key individuals.

P14L14: Why don't you use the current Real to Dollar exchange rate? The current minimum wage is R\$ 1,045 and the exchange rate is currently 5.3 Real per Dollar. Thus, minimum wage is approximately US\$ 197. I think you should use this current exchange rate (2020) for all the conversations of Real to Dollar you have made in the manuscript.

It makes sense to use the currency conversions relevant to this period (2016-2019) when these data were generated.

It sounds strange to me the way some of the coauthors are mentioned in the text. For instance, Laury is a coauthor in the paper but in many parts he is mentioned in a way it seems he is

outside it: P24L31: “Laury’s approach combined... This occurs with other coauthors too. I am a non-native English speaker. Not sure if it is acceptable or not in English.

This was tricky as Laury is part of the story. It is a narrative after all.

P20 – Paragraph starting L37. You mention a dream map and the idea of large forest corridors. Which is the goal in terms of forest cover and/or forest connectivity for the project’s landscape? Was it discussed or established at any moment as a goal? Which have the project achieved in terms of forest cover increase in the landscape so far?

This information was added: The Dream Map is an approach to landscape planning, created by IPÊ, and discussed with many stakeholders in the region during Eco-Negotiations, which are participatory meetings held at the Morro do Diabo State Park, or rarely in the public attorney headquarters in Presidente Prudente, the largest city in western São Paulo. The Dream Map is based on scientific assessments of water sources and rivers, the presence of endangered species in remaining habitats, along with information on who owns the land and where the priority areas for conservation are located. The goal is to maximize efforts on where to plant riparian forests, corridors and buffer zones, for example. All aspects are described publicly, so the understanding can lead to acceptance and social engagement in environmental issues. All of this information is pulled together to identify areas where reforestation efforts would be most beneficial and feasible. The dream map guided the creation of Brazil’s largest reforestation corridor system, which after ten years of effort, links two main remnants of Atlantic Forest in the Pontal de Paranapanema region, the Black Lion Tamarin Ecological Station and the Morro do Diabo State Park (Fig. 10).

The FLR process is not about adding hectares but about reversing drivers of degradation and deforestation and improving land use and tree cover. It will take a long time to see significant changes in forest cover in this landscape, but that does not mean that restoration is not happening.

P21L35 – You could additionally state the mean annual increment of CO₂ or carbon per hectare per year by corridors. Moreover, instead of saying since 2006, you should state the right period. Is it from 2006 to 2020? Finally, I suppose you are mentioning the gross CO₂ uptake by forest restoration. I would not state this is CO₂ equivalents neutralized if you are neither discounting baseline (previous land use prior to reforestation) nor emissions during forest restoration implementation. If you are considering both discounts, ok. But add a brief sentence to make it clear.

This section has been changed. It now reads: From 2012 to 2018, reforested corridors have grown over 1500 ha (Fig. 10) neutralizing a net estimate of 156,000 Mg of CO₂ equivalents (MgCO_{2e}) in the Pontal region after discounting for the baseline stocks of local pasturelands – where restoration is carried out – of 14.3 MgCO_{2e}/ha (36). Local data on restoration sites indicate an annual gain of 12.8 MgCO_{2e}/ha of restoration during the first five years of restoration, with a potential to reach a total stock of 317.2 MgCO_{2e}/ha after 30 years (37). By 2015 and 2017, a total of approximately 766 Mg of CO₂ equivalents were stored in aboveground biomass in the 122 hectares restored with support from WeForest (38).

Table 1: The caption could have more details and mention that Pontal do Paranapanema is in Brazil. Also, I think this table should not have horizontal lines, except in the borders. In addition, I think the deadline of each project could be added in case of projects that are no longer active. Also, a second column with project goal could be added. Finally, instead of only marking “X”, when available, you could put the goals and expected metrics in each activity for each project

Figure 1 has been modified and combined with previous Figure 2 and shows on top a simple map of the region within São Paulo State and Brazil. The goals of the project in Table 1 are described in the text. The table is intended to show the bigger picture of projects and interventions that expanded over time.

Key enabling factors and partnerships:

1 – highlight that dialogues between MST settlers and large landowners are frequently hard and not easy to be conducted. If that was the case at Pontal, it is worth mentioning to increase the importance of the IPÊ as a mediator for solving part of these conflicts.

Done

6 – Is the carbon credits somehow verified by a third part in these processes (example VCS)? Who is the “owner” of this carbon offset? IPÊ?

Yes, there are third parties involved in verification of carbon credits.

Figures: I think figures could be overall improved and, when possible, those that were obtained from other sources (especially the maps) could be substituted or combined with figures prepared by the authors. Also, I think 10 figures are too much.

Figure 1, 2 and 10: They are showing basically the same landscape and information. If I understood, figures 1 and 2 were not produced by the authors. I encourage you to produce your own figures for the manuscript. In that case, I suggest merging figures 1, 2 and 10 in a colorful and informative figure 1. Another option is to merge 1 and 2 in a new figure, and put this new figure together with figure 10, that would be a single figure with panel A and B. This would show respectively the current/past landscape and the dream landscape. The mention of the protected areas (better to say protected areas instead of conservation units, as in table 2 caption) in 2 could be in figure 1. Names of the protected areas should be described in the figure caption. What does “C” mean in figure 1? Is it necessary to mention? If not, remove them. If yes, explain in the caption.

Figure 1 has been replaced and combined with Figure 2. The map in Figure 7 is stylized and focuses on different activities, so they are not redundant. Two photos were combined into Figure 4. So there are now 7 figures.

Figure 4 maybe could be a supplemental material.

Previous Figure 4 has been moved to S2.

Figure 5: The caption is too long. This could possibly be in the text, in the end of the paragraph starting P13L31. In addition, there is no citation to figure 5 in the text.

Figure 3 (now) legend has been shortened. We wanted to keep the quote in there.

Figure 6: I suggest moving the citation of this figure in the text to the end of the sentence
“Agroforestry islands based on....and the park (Fig. 6).

Done

Figure 5-9. I suggest to put all of them in a single (or maybe two) figure with panels A-E.

Two photos are combined. If the editor wants more photos to be combined, we can do this.

Figure S1 – You should somehow indicate where the corridors are in the figure. At least, the most relevant ones should be pointed out.

The resolution will not permit the corridors to be seen. This is shown in Figure 7.

Minor comments

P4L41: (6,7)

Fixed

P5L12: provide authorship for *Leontopithecus chrysopygus*? If yes, do it for other names you mentioned.

Not done. None of the publications on this species provide authorship information.

P5L22: FLR program...

Done

P6L31: you mean the São Paulo state? Or use state with the general meaning of country or nation?

State of São Paulo

P6L36: remove Supplemental Materials

Done

P9L10: 1,000

Done

P9L40: unit of conservation = area?

No, we mean unit of conservation action

P10L12: use hectares instead of acres

Done

P12L43: change the “.” By “,” before “many aspects”.

This sentence has been changed

P12L41: Ribeirão?

Fixed

P14L15-16: to two minimum wages...

Done

P17L26: forest restoration plantings or ecological restoration tree seedling plantings? Just to be clear it is about tree seedlings planted to restore tropical forests.

This is a comparison between planted forests (seedlings planted) and agroforests.

Now use the term “restoration plantation of mixed native species” to be clear.

P21L35: 1,500

Fixed

P21L37: Mg instead of tonnes?

Changed to Mg

P21L41: Cite the WeForest report?

Done

P27L54: remove “22.”

This is part of the page number so I did not remove it